# Spatial Frequency Tuning of Body Inversion Effects

**DOI:** 10.3390/brainsci13020190

**Published:** 2023-01-23

**Authors:** Giulia D’Argenio, Alessandra Finisguerra, Cosimo Urgesi

**Affiliations:** 1PhD Program in Neural and Cognitive Sciences, Department of Life Sciences, University of Trieste, 34128 Trieste, Italy; 2Laboratory of Cognitive Neuroscience, Department of Languages and Literatures, Communication, Education and Society, University of Udine, 33100 Udine, Italy; 3Scientific Institute, IRCCS E. Medea, Pasian di Prato (Udine), 33037 Udine, Italy

**Keywords:** body inversion effect, body perception, spatial frequency, configural processing, parvocellular, magnocellular

## Abstract

Body inversion effects (BIEs) reflect the deployment of the configural processing of body stimuli. BIE modulates the activity of body-selective areas within both the dorsal and the ventral streams, which are tuned to low (LSF) or high spatial frequencies (HSF), respectively. The specific contribution of different bands to the configural processing of bodies along gender and posture dimensions, however, is still unclear. Seventy-two participants performed a delayed matching-to-sample paradigm in which upright and inverted bodies, differing for gender or posture, could be presented in their original intact form or in the LSF- or HSF-filtered version. In the gender discrimination task, participants’ performance was enhanced by the presentation of HSF images. Conversely, for the posture discrimination task, a better performance was shown for either HSF or LSF images. Importantly, comparing the amount of BIE across spatial-frequency conditions, we found greater BIEs for HSF than LSF images in both tasks, indicating that configural body processing may be better supported by HSF information, which will bias processing in the ventral stream areas. Finally, the exploitation of HSF information for the configural processing of body postures was lower in individuals with higher autistic traits, likely reflecting a stronger reliance on the local processing of body-part details.

## 1. Introduction

Classical models of object perception claim a separation of labor for the dorsal and ventral visual streams, with the first being involved in object motion processing and the latter in object identity recognition [1,2]. Notably, these two pathways are biased toward different spatial frequencies in the scene, which are conveyed by magnocellular or parvocellular signals and are better qualified to support the processing of large spatial differences or the analysis of local features, respectively [3,4,5,6]. Indeed, the anatomical mapping of magnocellular and parvocellular regions has revealed a preference in the dorsal pathway for low spatial frequency (LSF) information [3], which is more critical when people are asked to make spatial coordinate judgments regarding stimuli and, thus, process global configurations. Concurrently, high spatial frequency (HSF) information seems to dominate processing in the ventral stream, which is more efficiently exploited when the observers need to integrate fine details in order to categorize objects [4,5,6]. Accordingly, it has been shown that high-order areas of the dorsal stream are more sensitive to the processing of LSF images [7], while the ventral areas that are generally dedicated to object recognition, such as the ventral part of the lateral occipital cortex, respond more strongly to HSF than to LSF content [8].

Compared to identifying objects, the recognition of human faces and bodies engages special perceptual processes and neural mechanisms [9,10]. Indeed, faces and bodies seem to be preferentially processed using a configural strategy, for which the detection of specific spatial relationships among features is more relevant than the single features per se [11,12,13]. This tendency is further reflected by peculiar perceptual effects, such as the so-called “inversion effect”, which do not occur, at least to the same extent, for objects [13]. In the inversion effect, people exhibit an impaired recognition of faces and bodies when these stimuli are presented upside down, in comparison to when they are presented in their canonical orientation [14,15,16]. This effect shows that changing the spatial orientation of a face or a body with respect to their canonical representation heavily disrupts the processing of global configuration, even though both upright and inverted stimuli convey the same amount of local information. Indeed, inversion prevents the otherwise favored engagement of configural mechanisms by sparing only the inspection of single details [17,18,19,20].

With respect to the different spatial frequencies, previous studies have provided mixed evidence about the exploitation of LSF and HSF information in the configural processing of faces. On the one hand, some authors have observed that in both matching-to-sample and whole-part-matching paradigms, the configural processing of faces favors LSF images compared to HSF images [21,22]. On the other hand, other researchers have pointed out that configural processing occurs equally for LSF and HSF faces [23]. Importantly, even if the face inversion effect can be detected at a behavioral level using either HSF or LSF images [24,25], an electrophysiological marker of the early stage of face processing, namely, the N170 stage, has been found to be less affected by face inversion for HSF than LSF or broadband faces [26,27,28]. This may point to a dynamic coarse-to-fine mode of face processing, in which the early stages of the visual processing of faces mainly rely on LSF to detect face configuration, while HSF faces are later processed to identify local details [21]. In keeping with this model, Goffaux et al. [8] found that short or long presentations of faces led to a preference of face-selective areas in the occipitotemporal cortex to LSF or HSF, respectively.

Even less is known about the contribution of different visual scene bands to the configural processing of bodies. A coarse-to-fine mode of body processing, similar to that reported for faces [21], may point to the priority of LSF information over HSF information for configural body perception also. Accordingly, Canario and colleagues [29] showed that both face- and body-selective areas in the occipitotemporal cortex are more activated by LSF than HSF images of their preferred stimulus (i.e., a face or a body, respectively), thus hinting at a preferential involvement of configural processing, embedded in magnocellular channels. In a similar vein, the neural activation of body-selective occipitotemporal areas was reported to be greater in response to whole bodies than in response to individual body parts [30] or to a scrambled assembly of the same body parts [31], pointing to a preferential response to body configurations. Differently, a previous neurostimulation study [32] used broadband body stimuli and assessed the body inversion effects during the visual processing of upright and inverted postures after interferential stimulation of areas within the dorsal or the ventral streams, namely, the ventral premotor cortex and a body-selective occipitotemporal area (i.e., the extrastriate body area (EBA)), respectively. The results of this study showed that, while interference with the ventral premotor cortex reduced the accuracy in matching upright body postures, thus diminishing the body inversion effect, interference with EBA impaired the matching of inverted bodies and enhanced the body inversion effect. Based on that evidence, the authors proposed the existence of two independent cortical routes that are involved in human body processing: one route is deputed to the configural process of whole bodies and is housed in dorsal system areas (such as the ventral premotor cortex), while a second route specialized in the processing of local body features and is housed in ventral system areas (such as the EBA).

The discrepancies among previous studies in highlighting the involvement of ventral and dorsal stream areas in body configural processing can reflect the relative contribution of the two streams to the processing of different aspects of the body. Indeed, ventral areas seem to be more engaged when unchanging aspects of the body are processed, such as body shape, while dorsal areas come into play when an observer treats more changeable attributes, such as body postures and movements [33,34,35]. In line with the flexible usage hypothesis [36], the initial categorization of a stimulus may influence its early perception, so that identical visual inputs may prompt the preferential use of LSF or HSF information, according to the task. Although our perception may use LSF more efficiently for configural processing and HSF more efficiently for featural processing [37], both configural and featural sources of information can be differently obtained from broadband spatial scale ranges [23,37], according to the type of perceptual categorization that is required by the task. Face perception studies have, indeed, shown that different scales of frequencies can support different categorization tasks. For example, face identity has been shown to be better recognized at coarser scales [9], while HSF seems to be crucial for expression recognition [38]. Conversely, gender judgment regarding faces has not revealed the compelling dominance of either HSF or LSF, with studies supporting either a bias for LSF [39] or HSF information [40] or, finally, providing evidence for the contribution of both spatial scales [41,42,43].

Based on these findings, we selected gender and posture dimensions as features reflecting, respectively, the unchanging and dynamic aspects of bodies. We hypothesized that processing body gender, as a proxy to body shape, may rely more on the activity of the parvocellular system and be biased toward HSF information. In contrast, processing body posture, as a proxy to body dynamics, may rely more on the magnocellular system and be biased to LSF information. To test this hypothesis, we administered a delayed matching-to-sample paradigm, asking participants to report which of two probe bodies matched a previously presented body sample. The matching and the non-matching probe bodies could differ either for gender, while keeping the same posture, or for posture, while keeping the gender constant (“Task”). In different trials, the sample and probe stimuli could be presented as either upright or inverted (“Orientation”) and could contain all spatial frequencies or be filtered to retain only HSF or LSF bands (SF). This way, we aimed to test the relative contribution of LSF and HSF to the body inversion effects (BIEs) and, thus, to the configural processing of body gender and posture. We expected that the discrimination performance and BIEs should be higher for HSF (and intact) images than for LSF images when the task required the participant to discriminate body gender. Conversely, the opposite pattern should occur when the task required the participant to discriminate body postures, with better discrimination and greater BIEs for LSF (and intact) images than for HSF images. Furthermore, since previous studies showed a deficit of configural body processing [44] and a bias for the local rather than global processing of visual information [45,46] in individuals with autism, we also tested the reliance on LSF vs. HSF presentation for configural body processing in relation to the distribution of autistic traits in our non-clinical sample. We expected that individuals with higher autistic traits should rely less on LSF and more on HSF information for configural body processing, thus showing lower BIEs, particularly in the case of LSF presentations.

## 2. Methods and Materials

### 2.1. Participants

Seventy-two students (forty-five women, mean age = 24.3, standard deviation (SD) = 6.76 years) who were recruited at the University of Udine took part in the present study. Written informed consent was obtained from all participants. The procedures were approved by the local ethics committee (Institutional Review Board, Department of Languages and Literatures, Communication, Education and Society, University of Udine, Study Protocol CGPER-2019-12-09-03) on 9 December 2019 and conformed with the Helsinki Declaration. Participants were right-handed, as ascertained with a standard handedness inventory [47], and reported normal or corrected vision. They were unaware of the purpose of the experiment and a detailed debriefing was provided only after the whole experiment was completed.

The sample size (*N* = 72) required for our 2 × 3 × 2 repeated-measures ANOVA design (Task × SF × Orientation) was determined with G*power software (Heinrich-Heine-Universität Düsseldorf, Düsseldorf, Germany) [48], using the “as in SPSS” option. The expected effect size was set at f = 0.303m based on a previous study of SF modulation of the face inversion effect [24], with an alpha level at 0.05, and the desired power (1-beta) at 90%.

### 2.2. Stimuli

We used the Character Creator 3.0 software (Reallusion, San Francisco, CA, USA) to generate a set of virtual human-body stimuli. The stimuli were constructed by selecting four virtual-human models (two females and two males) from the software default database (i.e., F1 and F2, and M1 and M2). To increase the stimuli variation, we produced four different versions of each model by setting the percentage of gender traits at 80% or 100% and the amount of body fat at 0% or 10%. Thereby, we generated more or less masculine/feminine and fatter/thinner bodies, obtaining a set of 16 different bodies. Furthermore, each body was rendered in four different daily poses (e.g., standing, idle, walking, moving) selected from among the default postures available in Character Creator; these postures were selected on the basis of the validation results of a previous study [49]. Then, the images were imported into Adobe Photoshop CS6 (Adobe Systems, San Jose, CA, USA) to manipulate the SF content by using a Gaussian blur filter with a 6-pixel kernel for low-pass filtering (LSF), and with the high-pass filter application set to a radius of 0.9 pixels for the high-pass filtering (HSF). Thus, in addition to the intact image, we obtained HSF and LSF versions of each image, for a total of 192 different stimuli. All the grey-scale bodies were pictured from a frontal perspective and against a black background. For all images, the head, pectoral, and pelvic areas were blurred to mask the facial and primary sexual characteristics, while keeping enough morphological information to visually convey the sexual phenotype (Figure 1).

### 2.3. Procedure

The experiment was created with E-Prime software (version 2.0, Psychology Software Tools, Inc., Pittsburgh, PA, USA). Given the COVID-19 pandemic, participants were tested individually at their houses, using a laptop PC with the E-prime Subject Station software installed; they were advised to go to a quiet room and sit in front of their computer screen (refresh rate, 60 Hz) at a distance of 60 cm. The screen and stimulus resolutions were adjusted according to the E-prime settings, to ensure that the images covered an area of approximately 10° × 5° for all participants and for each type of stimulus. In two different sessions, participants were asked to perform two identical matching-to-sample tasks, which differed only for the identity of the models used as stimuli (F1 and M1 in one session, F2 and M2 in the other session). In each session, lasting approximately 40 min, participants were administered 8 blocks of 96 trials, with a random presentation of upright and inverted stimuli in the intact, LSF, and HSF conditions and requiring the discrimination of either gender or posture. Thus, across the two sessions, participants were administered a total of 1536 trials (128 trials per cell).

Each trial started with the appearance of a white central fixation cross (500 ms) presented on a black background, followed by the sample body form, lasting 150 ms, which was also presented on a black background. Soon after the image offset, a scrambled mask was presented for 500 ms. Then, participants were presented with two probes, positioned one next to the other to one side of the screen’s center, and were asked to identify the body image that matched the target. To give their responses, they used the left or right index finger to press the “Z” or “M” key on their QWERTY keyboard; each key corresponded to one of the two positions on the screen on which the probes were presented. The location of the matching and non-matching probes was counterbalanced across trials of the same participants, ensuring that, in half of the trials, the matching probe was presented on the left side of the screen, and in the other half, it was presented on the right side. The two probes remained on the screen either until a response was received or for a maximum of 3000 ms, after which a new trial was presented (Figure 2). The two probes could differ either for gender or for posture. Importantly, both the sample and probe stimuli could be shown in an upright or inverted orientation, with the orientation of the probes always being consistent with that of the sample. Participants were asked to give their responses as accurately and quickly as possible.

In addition, at the end of the experimental session, participants filled in a computerized version of the autistic quotient (AQ) model [50]. This is a 50-item self-report questionnaire that is largely used to estimate the presence of autistic traits in a non-clinical population. It measures impairments related to either cognitive functions (i.e., attention-switching, attention to detail, and imagination) or social skills (i.e., communication and social skills) that characterize the autistic profile [51].

### 2.4. Data Handling

Analyses were performed using an analysis of variance (ANOVA) design and the HSD Tukey post hoc test for multiple comparisons, implemented using the STATISTICA 7 software (StatSoft, Tulsa, OK, USA). Individual performance in the matching-to-sample task was expressed as a percentage of correct responses (Accuracy, %) and mean Reaction Times (RTs, in ms) of the correct responses for each individual and condition. Data from five participants, four of whom were women, were excluded due to their presenting an accuracy rate below 50% (*n* = 3) or due to technical issues (*n* = 2). Therefore, data from sixty-seven participants were entered into the analyses. Furthermore, from both the accuracy and RT analyses of the remaining participants, we excluded those trials that yielded results that were two SD values above or below the individual RTs mean, which represented 4.89% of the total trials. “Accuracy” and “RTs” were thus entered as dependent variables in two separate RM ANOVAs that had “Task” (2 levels), SF (3 levels), and “Orientation” (2 levels) as repeated measures

For each participant and condition, we calculated a BIE index for both Accuracy and RTs. To obtain the Accuracy BIE score, we subtracted the accuracy for inverted bodies from the accuracy for upright bodies that were within the same condition. To obtain the RT BIE score, we subtracted the mean RTs for upright bodies from the mean RTs for inverted bodies that were within the same condition. This way, for both Accuracy and RTs, the greater the BIEs are, the larger the drop in performance for inverted vs. upright bodies, and thus, the larger the use of configural body processing. We performed planned comparisons between the strengths of the BIEs across the three spatial frequency conditions, using a paired dependent sample *t*-test (two-tailed) with Bonferroni correction to control for multiple (6) comparisons. Furthermore, individual RT and Accuracy BIEs for each condition (i.e., gender—HSF, gender—LSF, gender—Intact, posture—HSF, posture—LSF, and posture—Intact) were correlated with the AQ total score. The significance of Pearson’s r coefficient was tested with a Bonferroni correction procedure to control for multiple (6) correlations. The significance threshold was set at *p* < 0.05 for all analyses. Effect sizes were estimated using partial eta-squared calculations (ηp2). Values are reported as the mean ± the standard error of the mean (SEM).

## 3. Results

### 3.1. Accuracy

The 2 Task × 3 SF × 2 Orientation repeated-measure ANOVA on Accuracy yielded a significant main effect of SF [F(2, 132) = 57.55; *p* < 0.001; η_p_^2^ = 0.466], which revealed that the body images were matched better for HSF (88.2 ± 0.7%) than for the intact image (86.1 ± 0.8%) and LSF (84.5 ± 0.8%; all *p* < 0.001) stimuli; also, performance was significantly lower for the LSF stimuli than for the intact images (*p* < 0.001). Further, a significant main effect of Orientation [F(1, 66) = 140.2; *p* < 0.001; η_p_^2^ = 0.679] emerged, confirming that discrimination between bodies was easier with the upright images (88.2 ± 0.8%) than the inverted images (84.3 ± 0.7%). Notably, the ANOVA showed significant Task × SF [F(2, 132) = 57.82; *p* < 0.001; η_p_^2^ = 0.467] and Task × Orientation interactions [F(1, 66) = 96.52; *p* < 0.001; η_p_^2^ = 0.594], further qualified by the three-way Task × SF × Orientation interaction [F(2, 132) = 4.45; *p* = 0.013; η_p_^2^ = 0.063]. All other effects were not significant (all F < 1.7, *p* > 0.19, η_p_^2^ < 0.02). The Tukey HSD post hoc test revealed that, for upright bodies, the gender task was performed better than the posture task in the Intact and HSF conditions (all were *p* < 0.001), while a better performance in the posture than in the gender task was seen for the LSF condition (*p* = 0.006). Conversely, for inverted bodies, the gender task was more difficult than the posture task in all SF conditions (all were *p* < 0.012). This pattern reflected a worsening of gender discrimination performance with either inversion or LSF manipulation.

Indeed, considering the modulations within the gender task for both upright and inverted bodies, the participants’ accuracy was significantly lower in the LSF than in the HSF and Intact conditions (all *p* < 0.001), which, in turn, did not differ (all were *p* > 0.48). Furthermore, upright body stimuli were matched more successfully than inverted body stimuli in all SF conditions (all *p* < 0.001; Figure 3a). Accordingly, with respect to the Accuracy BIE scores, planned *t*-tests comparing the three SF conditions (Intact: 7.69 ± 0.913%; HSF: 7.45 ± 0.552%; LSF: 6.73 ± 0.824%) did not reveal any differences (for all comparisons, t(66) < 1.1; Bonferroni-corrected *p* > 0.99).

Concerning the posture task, post hoc tests showed that upright bodies were matched less successfully when presented as intact images than with the HSF and LSF maniuplations (all *p* < 0.001), which, in turn, did not differ (*p* = 0.99). Conversely, performance regarding the inverted bodies was comparable across all SF conditions (all were *p* > 0.39). Furthermore, comparing the upright and inverted bodies showed no significant difference in any SF conditions (all *p* > 0.23 (see Figure 3c)). Similarly, planned *t*-tests on the Accuracy BIE revealed that the difference between upright and inverted bodies did not differ across SF conditions (Intact: −1.16 ± 0.913%; HSF: 1.15 ± 0.668%; LSF: 1.78 ± 0.661%; all t(66) < 2.166, Bonferroni-corrected *p* > 0.059).

### 3.2. Reaction Times (RTs)

The 2 Task × 3 SF × 2 Orientation repeated-measures ANOVA performed on the RTs showed a significant main effect of the Task [F(1, 66) = 29.80; *p* < 0.001; η_p_^2^ = 0.311], revealing that, overall, participants were more rapid in matching the bodies in terms of their gender (733.56 ± 15.35 ms) than for their posture (771.22 ± 15.61 ms). A main effect of SF was also found [F(2, 132) = 27.349; *p* < 0.001; η_p_^2^ = 0.293], which revealed that bodies were matched faster in the HSF (734.37 ± 14.49 ms) than in the Intact (754.94 ± 14.72 ms, *p* < 0.001) and LSF (767.86 ± 16.61 ms; *p* < 0.001) conditions; also, responses in the LSF condition were significantly slower than in the Intact condition (*p* < 0.01). Furthermore, a significant main effect of Orientation [F(1, 66) = 60.66; *p* < 0.001; η_p_^2^ = 0.478] was found, thus attesting that body recognition was faster for upright (727.97 ± 14.01 ms) rather than for inverted (776.82 ± 16.67 ms) images. Notably, the ANOVA conducted on the RTs also showed significant Task × SF [F(2, 132) = 20.70; *p* < 0.001; η_p_^2^ = 0.239], Task × Orientation [F(1, 66) = 127.69; *p* < 0.001; η_p_^2^ = 0.659], and SF × Orientation [F(2, 132) = 17.08; *p* < 0.001; η_p_^2^ = 0.205] interactions, all further qualified by the Task × SF × Orientation interactions [F(2, 132) = 19.99; *p* < 0.001; η_p_^2^ = 0.232]. The Tukey HSD test reported that, in all SF conditions, participants discriminated the gender of upright bodies more rapidly than their posture (all were *p* < 0.001), while no task difference was obtained in the case of inverted body images (all were *p* < 0.43).

Considering the modulations in the Gender task for upright body images, the participants’ performance was significantly slower for the LSF than the HSF and Intact stimuli (all *p* < 0.001), which, in turn, did not differ (*p* = 0.602). Similarly, for inverted body images, performance for the HSF stimuli was significantly faster than that for the LSF stimuli (*p* < 0.001) and tended to be faster than that for Intact stimuli (*p* = 0.062); RTs for the LSF and Intact stimuli did not differ (*p* = 0.938). Finally, upright body images were matched faster than inverted body images in all SF conditions (all *p* < 0.001 (see Figure 3b)). Planned comparisons of the drop of performance for the inverted compared to the upright body images across the three SF conditions revealed that the RT BIE was significantly smaller in the LSF condition (60.10 ± 84.79 ms) than in the Intact [107.63 ± 80.56 ms; t (66) = −5.97; Bonferroni-corrected, *p* < 0.001] and HSF [101.24 ± 77.29 ms; (t(66) = −4.51; Bonferroni-corrected, *p* < 0.001] ones, which in turn did not differ [t(66) = 0.85; Bonferroni-corrected, *p* > 0.9].

In terms of the Posture task, post hoc tests showed that the upright body images were matched faster for the HSF than for the LSF and Intact conditions (all *p* < 0.001), which, in turn, did not differ (*p* = 0.13), while the performance for inverted body images was comparable in the three SF conditions (all *p* > 0.99). Furthermore, only the performance for the HSF stimuli was better for upright than inverted body images (*p* < 0.001), while for the LSF and Intact conditions, the performance for upright and inverted body images was comparable (all *p* > 0.12 (see Figure 3d)). Planned *t*-tests on the RTs BIE scores showed that the difference between upright and inverted bodies was higher in the HSF (38.55 ± 7.29 ms) than in the Intact [−17.73 ± 8.99 ms; t(66) = 5.44; Bonferroni-corrected, *p* < 0.001] and LSF [3.31 ± 6.86 ms; t(66) = 3.93; Bonferroni-corrected, *p* = 0.001] conditions, which, in turn, did not differ [t(66) = −2.2; Bonferroni-corrected, *p* = 0.187].

### 3.3. Correlations

Concerning the correlations between the autistic traits and the BIE indexes for each SF and task condition, we only obtained a significant negative correlation between the AQ and the Accuracy BIE for HSF body images in the posture task [r(67) = −0.38; Bonferroni-corrected, *p* = 0.01]. This correlation suggested that participants with higher levels of autistic traits were less sensitive to BIE when detecting body posture from parvo-biased stimuli (Figure 4). No other correlations were significant (all were Bonferroni-corrected at *p* > 0.15).

## 4. Discussion

The present study aimed to test the contribution of HSF and LSF information to the configural processing of body gender and posture. To do so, we presented upright and inverted body images, either as broadband or filtered to HSF or LSF, and required the participants to discriminate bodies that differed in either gender or posture. Based on knowledge about the relative preference of parvocellular and magnocellular channels for, respectively, HSF and LSF information [3], along with their relative involvement in the processing of form or motion aspects of human bodies [35], respectively, we expected better performance and a greater BIE for the HSF body images than for the LSF body images in the gender discrimination task, and for the LSF body images than for the HSF body images in the posture discrimination task.

In keeping with these expectations, we found that filtering out HSF information from the body images disrupted the discrimination of body gender, with lower accuracy and slower RTs for LSF than for both intact and HSF stimuli. This preference for HSF is in line with the notion that perceiving body gender may require diagnostic information about morphology and the proportions among body parts [52,53,54]. These stable aspects of the body may be better conveyed through parvocellular than magnocellular channels [32,34,55]. Nevertheless, gender judgment regarding faces has not revealed a compelling dominance of HSF or LSF, with studies supporting a bias either for LSF [39] or for HSF information [40], or, finally, providing evidence for the contribution of both spatial scales [41,42,43]. The different contributions of HSF and LSF to gender perception from bodies and faces might depend on the specific type of gender-typing features. Here, we used three-dimensional body renderings, and, even if primary sexually dimorphic cues (i.e., breasts and genitals) were blurred, male and female bodies differed not only in their global proportions (i.e., width of shoulder, torso, and hip, and their relationships, such as the waist-to-hip ratio) but also for their internal secondary sexually dimorphic cues (i.e., musculature, body-fat distribution), which may characterize gender perceptions from either bodies or faces [56]. This might have facilitated the discrimination of gender, based on the HSF information in our study and in previous studies that found that facial gender perception is better supported by HSF [40].

A partially different pattern of findings was obtained for the posture discrimination task, with improved performance after filtering the images to keep mostly either HSF or LSF information. Indeed, the discrimination of upright body posture was facilitated more for HSF than intact images in terms of both accuracy and speed of response. In a slightly different way, the discrimination of LSF upright postures was facilitated in terms of accuracy, but not of the speed of response, compared to the discrimination of intact images. No effects of SF, instead, were obtained for inverted body postures.

The different effects of SF manipulation for the gender and posture discrimination tasks extend to body-processing the validity of the flexible usage hypothesis [36] by documenting a change in preference for different sources of information, depending on the types of body features relevant for each task. However, in contrast with our expectations, body posture discrimination information was not mainly contributed by LSF, instead involving both HSF and LSF sources of information. This finding contrasts with the prominent role of the magnocellular channel in the processing of dynamic features of the body [35], and also supports the contribution of the parvo-cellular channel in encoding body shape and posture and integrating them into a perceptual framework [57]. In this way, the joint contribution of LSF and HSF to body posture perception might reflect the coarse-to-fine temporal dynamics of LSF and HSF processing in ventral body-selective areas, as is proposed for faces [21]. Indeed, the precedence of LSF over HSF information may reflect the fast- vs. slow-acting feedback connections that operate within primary and secondary visual cortices during the visual perception of a stimulus [17]. Thus, the manipulation of the two frequency bands may affect the correctness of the final response or its speed differently [58], possibly explaining why the accuracy and the speed of responses were affected differently by the removal of HSF and LSF information in our study. A coarse-to-fine/fast-to-slow progression of the contribution of the magno- and parvocellular pathways has been proposed for general object recognition [59]. According to this model, LSFs are quickly extracted and conveyed by the magnocellular pathway, in order to trigger preliminary predictions regarding object identity; at this point, HSFs would take part in the process, activating the matching visual representations within the parvocellular system [60]. This model has been recently extended to include body action perception [61], based on evidence showing that LSF-based, magnocellular information about actions may not only be processed in the dorsal action observation network but may also exert feedback impact on later processing via fast connections between the occipital and prefrontal cortexes in both ventral and dorsal areas. These fast and slow connections may support the mixed contribution of both HSF and LSF regarding body posture discrimination, as seen in our study.

The flexible usage hypothesis [36] may also explain the unexpected finding that matching intact upright body posture was worse than matching HSF (for both Accuracy and RTs) and LSF (only for Accuracy) upright stimuli. Indeed, assuming the additive processing of different spatial bands, in which each set of source information provides fixed cues to a given task, intact images are expected to be always processed better, or at least equally well, than filtered images. However, the visual system may use different spatial frequency bands flexibly to solve a task, depending on the available information and the top-down control of the handling task [58]. Indeed, studies on face perception have shown that better performance in matching two stimuli was obtained when both stimuli were filtered to the same SF band, compared to when the two stimuli contained broadband or, even more noticeably, different SF bands [62,63]. This shows that, at least in certain tasks, limiting the processing of complex stimuli (such as faces and bodies) to given sources of information, as conveyed by one of the SF channels, might be advantageous. Similarly, HSF and LSF bands may convey different pieces of information about body posture, for example, regarding articulation joints or the global displacement of body parts in space, respectively. The parallel processing of these different aspects of body posture in intact images may thus complicate the matching-to-sample decision, leading to better performance for filtered than for intact images. A similar advantage for filtered rather than intact displays of ambiguous actions has recently been reported in a study [61] that aimed to investigate the contribution of different cues to action understanding, namely, body kinematics and contextual objects. These cues were thought to rely mainly on the different SF channels, with body kinematics conveyed mainly by LSF and object information conveyed by HSF. Accordingly, when kinematics and contextual objects provided incongruent cues to action understanding, removing one source of information by filtering the images to HSF or LSF images improved the participants’ performance, compared to the processing of intact stimuli.

While the discrimination of the gender and of the posture of upright body stimuli relied on HSF and LSF information differently, the results emerging from the BIE analysis suggested that configural body processing was better supported by HSF in both tasks. Indeed, when we compared the amount of performance drop for the inverted rather than upright stimuli across the SF conditions of the gender task, we found a reduction in the RT BIE, but not in the Accuracy BIE, for the LSF stimuli, compared to both the HSF and intact stimuli. Still, a significant body inversion effect was found for all SF conditions, suggesting that the configural processing of body gender involves both HSF and LSF information; however, there is a preference for HSF. Notably, the BIE for posture discrimination was less reliable than the BIE for gender discrimination; it only reached significance in the HSF condition. Indeed, in the posture task, reliable BIEs, which were again limited to RTs, were found for the HSF only and not for the LSF or intact images. To summarize, in the gender task, removing the HSF information affected gender discrimination because it slowed down the configural body processing when only LSF information was available. In a similar vein, in the posture task, limiting the visual information to HSF facilitated the configural processing of body posture, while no evidence of configural processing was found for LSF (and intact) images. Thus, although the impact of SF filtering on the amount of configural body processing was different for the two tasks, configural processing was greater when only HSF rather than only LSF images were presented, in terms of both gender and posture discriminations. This might suggest that configural body processing is better supported by parvo-biased, HSF information.

The preference for body configural processing with HSF seems to diverge from the dynamic coarse-to-fine mode of processing that is proposed for faces, which should prioritize the configural processing of LSF [21,23]. However, other studies have questioned this notion by documenting comparable face inversion effects across different spatial frequencies [24,25]. Furthermore, although human faces and body shapes are both biological stimuli that are relevant for social communication, their visual structure is inherently different and may trigger configural processing at different levels. In this regard, several authors have described a configural processing continuum in which different types of processing mechanisms can be distinguished, starting from part-based processing up to the holistic processing of the whole stimulus [16,63]. As addressed by Minnebusch and Daum [64], mechanisms for face and body perception may share the earlier stages of this continuum, namely, first-order relational information and structural information, while dissociations might take place at later stages, at the level of holistic processing. Accordingly, while studies converge in showing comparable inversion effects, and, thus, configural processing for faces and bodies [15,16], there is inconsistent evidence for holistic body processing [65]. This suggests that body forms might not be processed as integrated representations to the same extent that faces are. It is worth noting that the type of configural processing that is indexed by BIE seems to reflect the processing of the first-order spatial relationships between different body parts within the general body structure (e.g., that the arms are above the legs) as well the representation of the general symmetric structure of a body (e.g., two arms and two legs that are connected to the torso) [16]. Indeed, while BIE has been reported for either whole or left and right body halves, which preserve the general body structure, no BIE has been reported for either scrambled or top/bottom body halves [16]. These aspects of body shapes may differently serve the discrimination of body gender and posture, thereby explaining the different contributions of HSF and LSF to the configural body processing of gender and posture.

In the same vein, there is evidence supporting the view that configural representations of face and body may differ in the hierarchy of representational space. Harris and colleagues [66] showed that bodies might be represented in a configural way, not only at the level of the whole body structure but also in terms of the finer-scale organization of basic parts (e.g., the arm), thus resulting in multiple configural processing tasks of the different components of the body schema. From this viewpoint, we may argue that HSF information may be a more appropriate way to elicit configural processing at the level of the within-part organization of body structure.

A further noteworthy result came from the correlation between the AQ score and BIE. We found that participants with a higher level of autistic traits showed lower Accuracy BIEs when asked to make posture discrimination judgments of HSF bodies. Albeit at an explorative level, this result might suggest that autistic traits are associated with a weak exploitation of HSF information for the configural processing of body postures. In fact, participants with the highest autistic traits were facilitated in terms of the local processing of inverted postures, compared to the configural processing of upright bodies (i.e., negative BIE scores), when dealing with HSF images. This is in line with previous studies that have tested BIE in autism and have provided evidence of the weak configural processing of body postures [44]. This impairment in configural body processing could be at least partially ascribed to a deficit in interpreting and emulating body movements [67]. Accordingly, individuals with autism are characterized by a cognitive style that is biased toward local details at the expense of the global processing of visual information [45,46]; they seem to gain an advantage in processing stimuli containing HSF information [61,68]. Together, data coming from previous research and the correlation that we observed in the current study converge to suggest that people with high autistic traits may adopt distinctive strategies for visual body processing, with a bias toward exploiting HSF for the local processing of body part details.

The conclusions that we can draw from this study need to be weighted in light of its limitations. First, we did not take into account the temporal dynamics that may characterize SF processing [8]; the relatively long presentation time (i.e., 150 ms) that we used might have hindered the earlier contribution of LSF. According to past research [8], the preference of the occipitotemporal areas for LSF peaks at 75 ms and decays at 150 ms after stimulus presentation. Hence, even if the stimulus duration was shortened in our study, with respect to previous studies [69], it could still be too long in the context of unveiling the contribution of LSF information. Thus, future studies need to test whether a greater contribution of LSF might be detected at earlier stages of stimulus processing by reducing the presentation timing. Additionally, the type of stimuli that we used, namely, static body representations of different postures, can only partially define the influence of SF on motion processing, since dynamic stimuli may elicit a greater contribution from the magnocellular system. Thus, future studies exploring body motion perception should include the presentation of moving stimuli (i.e., videos) to clarify to what extent the bias of configural processing for HSF information in our study was due to the presentation of implied rather than actual movements. Finally, although the dorsal and the ventral pathways preferentially receive inputs from the magnocellular and parvocellular systems, respectively, our behavioral results cannot inform on the involvement of either one of the two routes. Indeed, the parvo- and magnocellular channels are not completely segregated in the cortex and there is evidence of inter-transmission between them in visual processing [70]. Future studies might need to combine this behavioral paradigm with electrophysiological, neuroimaging, or neurostimulation techniques, to explore the involvement of the ventral and dorsal areas in the processing of HSF and LSF information about body gender and posture.

## Figures and Tables

**Figure 1 brainsci-13-00190-f001:**
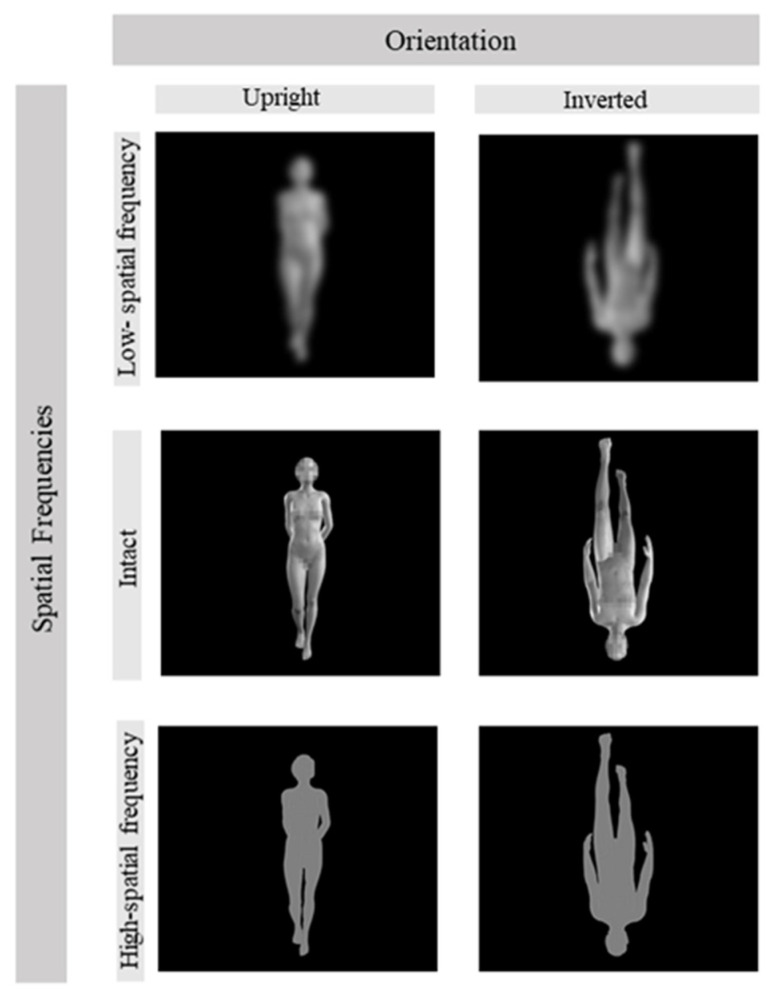
Examples of female and male virtual models used in the study as stimuli. The figure depicts the variation in orientation (upright vs. inverted) and spatial frequency (high spatial frequency, vs. intact images, and vs. low spatial frequency).

**Figure 2 brainsci-13-00190-f002:**
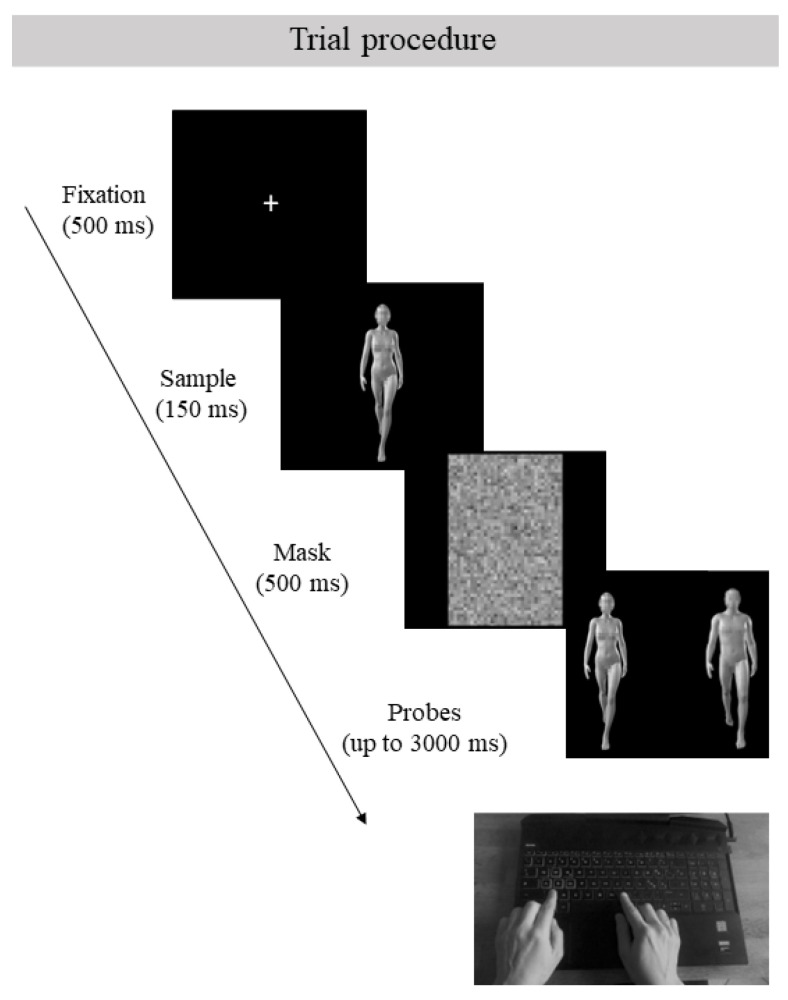
Trial procedure. A representation of the events constituting the structure of a single trial of the matching-to-sample task. Each trial began with the fixation cross, followed by the target picture. Soon after the mask appeared, a frame with the two probes remained on the screen until a response was recorded or for a maximum of 3000 ms. The background color of the screen remained black for the whole duration of the experiment.

**Figure 3 brainsci-13-00190-f003:**
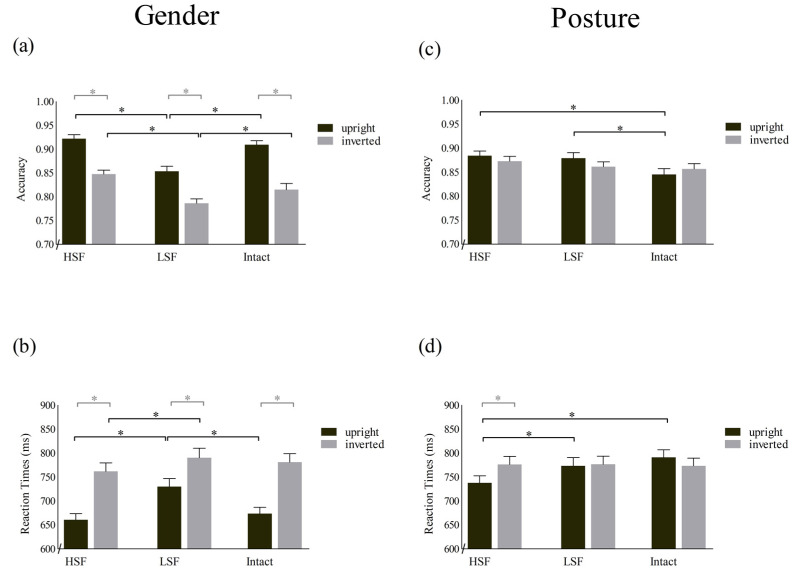
Participants’ performance in the matching-to-sample task under the different spatial frequency conditions (i.e., intact, LSF, and HSF) of upright and inverted bodies. Means and standard errors of the mean for “Accuracy” and “Reaction Times” (RTs) in the Gender task are reported in panels (**a**,**b**), respectively. Means and the standard error of the mean for “Accuracy” and RTs in the Posture task are reported in panels (**c**,**d**), respectively. Asterisks indicate a significant comparison (*p* < 0.05) between the SF condition (black brackets) and orientation (grey brackets).

**Figure 4 brainsci-13-00190-f004:**
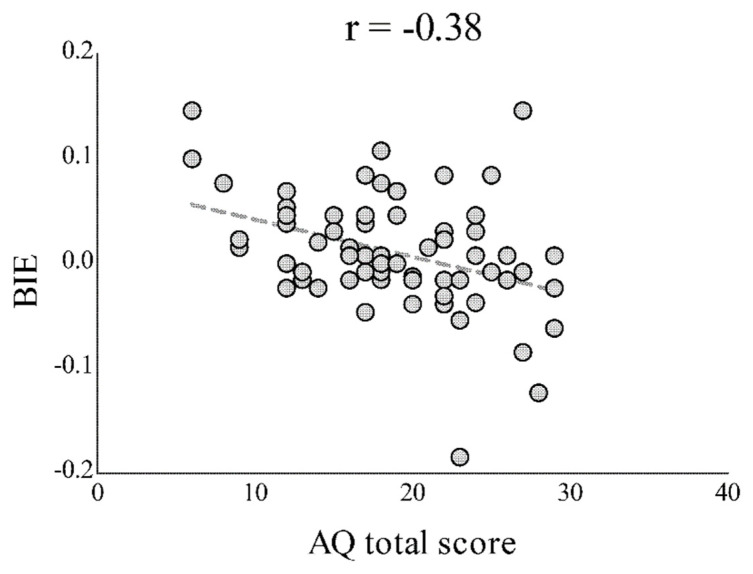
Correlation between the body inversion effect (BIE), calculated as the difference between the accuracy rating for upright body images and the accuracy rating for inverted body images, and the total score for the autistic quotient (AQ). This relationship indicates that the higher the participant’s level of autistic traits, the lower the difference between the posture recognition ability of upright and inverted high-filtered body images.

## Data Availability

The datasets generated and analyzed during the current study are available from the Open Science Framework, at this link: https://osf.io/8f3t2/. The E-prime code used during the current study is available at https://osf.io/8f3t2/.

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
