# Peer review of "Spatial Frequency Tuning of Body Inversion Effects"

_brainsci, 2023, doi:10.3390/brainsci13020190_

Round 1
Reviewer 1 Report
The paper addresses an interesting issue, with a proper experimental design and statistical analyses. Discussions are thorough and consider limitations to the present study.
I have just two requests of clarifications.
1. The authors should make clearer the discussions on BIEs. Indeed, the overall interpretation is that “configural body processing was better supported by HSF in both tasks“ (line 392). This is not of immediate comprehension to the reader.
The authors reach this conclusion because they interpret results on HSF images (RTs in the Posture task) as due to the presence of high spatial frequency information (thus considering HSP as fundamental to the task) and results on LSF images (ACC and RTs in the Gender task) as due to the absence of high spatial frequency information (thus considering HSF as fundamental to the task). I acknowledge that results on gender task are based on the fact that HSF and Intact conditions show better performance than LSF condition and they do not differ between each other. However, these results can also be interpreted as the presence of only low spatial frequency information worsen performance in upright stimuli (i.e., reduces accuracy and increases reaction times with respect to HSF and Intact conditions).
Also, interpretation of RTs BIE results (Gender Discrimination task: LSF < HSF and Intact; Posture Discrimination task: HSF > LSF and Intact) is given in terms of slowed or facilitated configural processing. However, differences are due to a change in RTs between the different upright conditions. The RTs never differ among the inverted conditions. The authors could clarify that the interpretation in terms of facilitation/slowing of configural processing is given because a significant difference between upright and inverted images was found within the LSF (gender task) and HSF (posture task) conditions.
As related to the point above, please clarify the following sentence: “In sum, greater configural processing was found for HSF than LSF images in both gender and posture discriminations, suggesting that configural body processing is better supported by parvo-biased, HSF information.” No greater configural processing was found for HSF images in gender discrimination task.
2. I find the paper lacks a comment on results on intact images in the posture task. Indeed, the authors should comment on why intact upright images show worse ACC than images limited to high or slow spatial frequencies.
Reviewer 2 Report
The aim of the current study was to test the contribution of high spatial frequency (HSF) and
low spatial frequency (LSF) information to the configural processing of body gender and posture.
Unfortunately, this manuscript needs substantial improvements and corrections before publishing may be possible.
General points:
Please add a list of abbreviations before References section to your manuscript.
Please check all spaces between the words and references numbers at the end of each sentences.
Special points:
Introduction
Lines 29-32: please add multiple references at the end of this sentence.
Lines 54-56: please describe exactly all these sentences.
Lines 101-103: please describe exactly all these sentences.
Methods
Lines 131-138: please add also the exactly date of the permission for all your experiments.
Figure 4: please improve the quality of this Figure.
Round 2
Reviewer 2 Report
Thank you for all corrections.